# T Cell Subsets and Natural Killer Cells in the Pathogenesis of Nonalcoholic Fatty Liver Disease

**DOI:** 10.3390/ijms222212190

**Published:** 2021-11-11

**Authors:** Yoseph Asmelash Gebru, Haripriya Gupta, Hyeong Seop Kim, Jung A. Eom, Goo Hyun Kwon, Eunju Park, Jin-Ju Jeong, Sung-Min Won, Satya Priya Sharma, Raja Ganesan, Dong Joon Kim, Ki Tae Suk

**Affiliations:** Institute for Liver and Digestive Diseases, Hallym University, Chuncheon 24252, Korea; yoseph@hallym.ac.kr (Y.A.G.); phr.haripriya13@gmail.com (H.G.); kimhs2425@gmail.com (H.S.K.); eomjunga32@naver.com (J.A.E.); ninetjd@naver.com (G.H.K.); epark312@hallym.ac.kr (E.P.); jj_jeong@hallym.ac.kr (J.-J.J.); lionbanana@hallym.ac.kr (S.-M.W.); satyapriya83@gmail.com (S.P.S.); vraja.ganesan@gmail.com (R.G.); djkim@hallym.ac.kr (D.J.K.)

**Keywords:** antigen presenting cell, T cells, NK cells, MHC I, MHC II, nonalcoholic fatty liver disease, gut microbiota

## Abstract

Nonalcoholic fatty liver disease (NAFLD) is a condition characterized by hepatic accumulation of excess lipids. T cells are commonly classified into various subsets based on their surface markers including T cell receptors, type of antigen presentation and pathophysiological functions. Several studies have implicated various T cell subsets and natural killer (NK) cells in the progression of NAFLD. While NK cells are mainly components of the innate hepatic immune system, the majority of T cell subsets can be part of both the adaptive and innate systems. Several studies have reported that various stages of NAFLD are accompanied by the accumulation of distinct T cell subsets and NK cells with different functions and phenotypes observed usually resulting in proinflammatory effects. More importantly, the overall stimulation of the intrahepatic T cell subsets is directly influenced by the homeostasis of the gut microbiota. Similarly, NK cells have been found to accumulate in the liver in response to pathogens and tumors. In this review, we discussed the nature and pathophysiological roles of T cell subsets including γδ T cells, NKT cells, Mucosal-associated invariant T (MAIT) cells as well as NK cells in NAFLD.

## 1. Introduction

Non-alcoholic fatty liver disease (NAFLD) is a liver impairment characterized by the presence of excessive hepatic fat accumulation caused by factors mainly related to a high fat diet. Moreover, existing data support that type 2 diabetes (T2D), hyperthyroidism and obstructive sleep apnea are important contributing factors for the progression of NAFLD [1,2,3]. A recent retrospective cohort study of women diagnosed with polycystic ovarian syndrome (PCOS) also reported that PCOS is a risk factor associated with subsequent NAFLD occurrence [4]. Its progression comprises a wide spectrum, ranging from simple hepatic steatosis to fibrosis and in extreme cases extending up to hepatocellular carcinoma (HCC). It comes with all manifestations and complications in liver impairment and is one of the major diseases with a global prevalence rate of 25% [5]. While the liver is mainly a large metabolic organ, it is also the first line of interaction between the host and external environment which makes it one of the primary targets for immunological responses. Considering its structural organization has profound implications for the circulatory system, the liver is persistently exposed to blood-borne microbes, mainly derived from the gut. Therefore, the liver is one of the most important immunological organs with several cells from both the innate and adaptive immune systems involved [6,7]. In NAFLD, a variety of innate and adaptive immune cells including Kupffer cells, neutrophils, T cells and natural killer (NK) cells have been indicated to be involved in the progression of hepatic steatosis, inflammation, and fibrosis associated [8,9,10]. These complex repertoires of non-parenchymal cells originating from both the lymphoid and non-lymphoid sources play key roles in hepatic immunoregulation and defense mechanisms.

T cells and NK cells are among the most common hepatic immune cells that are distinct in frequency and phenotypes depending on the progression stages of NAFLD [11,12]. During the earlier stages of NAFLD, a high-fat diet is usually associated with alterations in the composition of gut microbiota, changes in the metabolically active hepatic tissue, such as fatty acid accumulation and oxidative stress which usually occurs in parallel. Here, fat accumulation and dysregulation of the microbiome are the key participators in guiding the subsequent functions and phenotypes of T cells and NK cells. The altered microbiome stimulates the immune response through biosynthesis and hepatic translocation of activating ligands while the fatty environment alerts and induces infiltration of immune cells by increasing the production of chemically reactive radicals. Therefore, immune cell recruitment and expansion are mainly induced by microbial products, such as peptides and metabolites from the gut and chemotaxis molecules, such as chemokine C-C motif ligand secreted from the fatty liver microenvironment. While the association of the most common classes of immune cells with several cardiovascular and autoimmune diseases is relatively well studied, the ambivalent roles of different subsets of intrahepatic T cells and NK cells remain unexplained. In this review, we describe the nature of T cell subsets and NK cells, as well as discuss their roles in the pathogenesis mechanisms of NAFLD comprising the gut-liver axis.

## 2. T Cells as Part of the Innate and Adaptive Hepatic Immune System

While the liver was biased for being unresponsive to the immune system, it is now well established that it is capable of considerable immune responses that produce pathological and protective impacts. Currently, it is well established that both the innate and adaptive immune systems are involved in the pathophysiology of the liver. In NAFLD, cells from each type of immune system alter liver physiology with different pathological characteristics and cell phenotypes observed at different progression stages [13]. With respect to the innate immune system, the most important players are Kupffer cells and monocyte-derived macrophages [14]. During the earlier development of NAFLD which is mainly characterized by simple steatosis, these cells secrete chemotaxis stimulating molecules, such as C-C motif ligand 2 (CCL2) which results in hepatic accumulation of macrophages originating from circulating monocytes [15,16]. This will then result in the production of large amounts of proinflammatory cytokines, including interleukin (IL)-1β and tumor necrosis factor (TNF) α which in turn accelerate hepatic steatosis, inflammation, and fibrosis. In addition to macrophages, NK cells and dendritic cells (DC) also contribute to progression of NAFLD as part of the innate immune system.

When it comes to the adaptive immune system, T lymphocytes are particularly responsible for the cellular immunity that is mediated by infected cells and subsequent effector pathways, while B lymphocytes constitute the humoral component which is mediated by the production and secretion of antibodies. The cellular component of the adaptive immune response is stimulated through the identification of antigens presented on major histocompatibility complex (MHC) class I or II molecules on antigen-presenting cells (APCs) (Figure 1) [17]. For classical MHC classification, any cell that can recognize a foreign antigen, break it into peptides, secrete the MHC molecule-antigenic peptide complex into the cell membrane and interact with T cells is an APC. This interaction is facilitated through specialized molecules known as T-cell receptors (TCR) on the surface of T cells together with the co-receptor cluster of differentiation (CD) 3 [18]. TCRs are predominantly composed of α and β subunits capable of undergoing random rearrangements of V (variable), D (diversity), and J (joining) gene segments depending on the specific T cell subset enabling specific recognition of a broad range of antigens [19]. Therefore, T cells can express TCRs capable of recognizing diverse antigens from pathogens, tumors, and the external environment while maintaining memory and self-tolerance [20].

### Classification of T Cells

The presence of different types of protein subunits in TCRs is used for the characterization and classification of T cells as αβ T cells and γδ T cells. While both types of T cells develop from multipotent progenitor CD4^−^–CD8^−^ double-negative thymocytes in the thymus, about 95% of T cells in human peripheral blood express the αβTCR while the remaining 5% express the γδTCR [21]. As all T cells express the CD3 co-receptor, it is used as a specific marker during identification and characterization experiments. In addition to the classification of T cells based on their TCR type, they are commonly classified into several subsets according to other characteristics including the type of special co-receptors usually used as surface markers, physiological functions, way of antigen presentation as well as non-conventional characteristics. The most famous subsets are CD4^+^ (T helper) cells which recognize MHC class II molecules and CD8^+^ (cytotoxic) T cells that recognize MHC class I molecules (Table 1). CD4 cells are mature T cells expressing CD4 glycoprotein on their surface while CD8 cells express CD8 glycoprotein. These glycoproteins are key co-receptors that facilitate T cell activation by enhancing the binding of TCR to the MHC-peptide antigen complex [22,23]. NKT cells are non-conventional T cells that recognize lipid antigens presented by the MHC-like molecule CD1d rather than peptide antigens. These cells also express the NK marker that makes them innate-like. Mucosal-associated invariant T (MAIT) cells are another CD8^+^ non-conventional T cells recognizing vitamin B2 metabolites presented by major histocompatibility complex class I-related (MR1) molecules on APCs [24,25]. In this section, the roles of the above T cell subsets in different stages of NAFLD progression are discussed.

## 3. Roles of Different T Cell Subsets in Nonalcoholic Fatty Liver Disease (NAFLD)

### 3.1. γδ T Cells

Γδ T cells can be considered as a bridge linking the innate and adaptive immune systems. On one hand, their TCR can recognize conserved structures on pathogens and host cells including non-peptide metabolites and heat shock proteins. On the other hand, they are double negative (express neither CD4 nor CD8 co-receptors) which is of innate nature and can recognize a lot of non-peptide ligands without the need for antigen presentation by MHC molecules [26]. The fact that γδ T cells can be activated without TCR restriction allows them to produce an immune response much faster than αβ T cells. Additionally, these cells have been found to be more enriched in the liver than in blood which makes them interesting targets for a contribution to liver pathogenesis [27]. γδ T cells have been reported to constitute about 2–10% of total T lymphocytes in the blood while they account for 15–25% of the total hepatic T cell population [28]. Depending on disease types and progression stages, γδ T cells may have pathogenic or protective roles in the liver. During NAFLD, they are recruited to the liver and have been implicated in exacerbating steatohepatitis through regulating CD4^+^ T cells and increased expression of interleukin (IL)17 [29,30]. Equipped with pattern recognition patterns (PRRs), such as toll-like receptors (TLRs) that can recognize microbial antigens, γδ T cells are suggested to play central roles in promoting intrahepatic inflammation [31,32]. Therefore, the accumulation of these cells in the liver most likely aggravates NAFLD progression [33]. A study that investigated the roles of γδ T cells in a high-fat diet-induced (HFD) induced NAFLD mouse model reported that they are major providers of IL-17A, a key pro-inflammatory cytokine in the liver [34]. These cells are enriched in HFD induced NAFLD models as well as NAFLD patients and their depletion was able to protect mice from the progression of NAFLD. More, interestingly, it was shown that the gut microbiota maintains the homeostasis of liver-resident γδT-17 cells in a lipid antigen/CD1d-dependent manner [34].

### 3.2. Cluster of Differentiation (CD)4^+^ T Cells

#### 3.2.1. CD4^+^ T Helper Cells

As discussed above, T cells undergo proliferation upon the first encounter of antigens in the lymph nodes and acquire further differentiation markers which guide them for distinct functions. The CD4^+^ T cells are responsible for the generation, maintenance and regulation of other immune cells through the production of cytokines, therefore, they are termed helper cells [35]. Based on their functional capabilities and cytokines production, T helper cells are subdivided into type 1 (Th1) and type 2 (Th2). Th1 cells are usually associated with inflammation and inducing cell-mediated immune responses through the secretion of proinflammatory cytokines including interferon (IFN) γ, IL-2, tumor necrosis factor (TNF) α, IL-6, IL-8 and IL-1β. On the contrary, Th2 cells produce anti-inflammatory cytokines, such as IL-1ra, IL-4, IL-5 and IL-10 that induce a humoral immune response by helping cells, such as B cells with antibody production [36,37]. CD4 cells have been mainly associated with pathologic roles during NAFLD including in fibrosis and hepatocellular carcinoma (HCC). A recent study that explored the effect of CD4 cells in HFD induced NAFLD in humanized mice found out that they promote the development of fibrosis. In this in vivo model, increased infiltration of CD4 T cells was observed, accompanied by an elevation in pro-inflammatory cytokines mainly IFNγ, IL-6, IL-17A, and IL-18 [33]. While depleting CD4 cells in the HFD mice models could not prevent hepatic steatosis, it significantly reduced liver immune infiltration and fibrosis. This suggests that CD4 cells promote NAFLD-induced inflammation and progression to fibrosis. In a different perspective, dysregulation of lipid metabolism during NAFLD induces a selective loss in intrahepatic CD4 T cells which in turn results in HCC [38]. Therefore, crosstalk between inflammatory signaling and lipid metabolism in the liver may elucidate the mechanisms of roles of CD4 T cells in NAFLD progression.

#### 3.2.2. CD4^+^ Regulatory T Cells

Another subset of T cells that account for 5–10% of the total number of CD4^+^ T cells in healthy humans are regulatory T cells (Tregs). As a T cell subpopulation, they are produced in the thymus and express the transcription factor forkhead box P3 (Foxp3) of the forkhead family [39]. Their importance in the development and pathogenesis of NAFLD is poorly recognized. While little is known about the significance of intrahepatic Tregs in NASH, it is well established that they support the homeostasis of immune response by inhibiting proliferation and activation of cytotoxic T cells [40,41]. Tregs have been confirmed to be involved in the prevention of autoreactive cell proliferation in autoimmune hepatitis, and a negative control of the immune response in NASH and organ tolerance during transplantation [42]. The induction of Tregs has been demonstrated to alleviate insulin resistance thereby contributing to the amelioration of liver damage. A study that investigated the effect of inducing Tregs using the immunosuppressant OKT3 in NASH patients found that immune parameters and liver function were improved [43]. Another recent study that found increased intrahepatic Tregs in mice NASH models also reported their detrimental roles [44]. However, further studies are required to elucidate the detailed mechanisms for their roles which can help their applications in the immunotherapy of NASH.

### 3.3. CD8^+^ Cytotoxic T Cells

The transition from hepatic steatosis, the simplest form of NAFLD to nonalcoholic steatohepatitis (NASH) is the key factor for the development of fibrosis and HCC [45,46]. A recent study that used a choline deficient high-fat diet has demonstrated that the progression of hepatic inflammation and transition to NASH is accelerated by CD8^+^ cytotoxic T cells [47]. A study that investigated obesity-related NAFLD found that CD8^+^ T cells regulate obesity and hyperlipidemia-associated NASH through the production of cytokines, such as IL-10 and TNFα, which drive the recruitment of macrophages and activation of hepatic stellate cells (HSCs) [48]. HSCs are pericytes that reside in the liver sinusoid in a quiescent state but serve as the major storage site for lipid droplets during a high-fat diet. As the NAFLD condition progresses, these cells release their lipid droplets, differentiate into myofibroblast-like cells and become activated, expressing α-smooth muscle actin (α-SMA) and secreting large quantities of extracellular matrix proteins which results in fibrosis. CD8^+^ T cells and α-SMA have also been reported to be elevated during the progression of NAFLD in humans. Moreover, CD8^+^ tissue-resident memory T cells have been reported to promote liver fibrosis resolution by inducing the apoptosis of HSC indicating their important roles in fibrosis resolution [49]. On the contrary, cytotoxic T cells have been implicated for exerting protective effects where they were able to restrict hepatic inflammation in mice with NASH accompanied by decreased plasma levels of liver enzymes. Therefore, further exploration is required to reveal the differential roles of CD8^+^ cytotoxic T cells in different conditions of NAFLD [50]. Additionally, the underlying mechanisms by which these immune cells are stimulated to infiltrate and proliferate in the liver are also yet to be explored especially from the gut liver axis perspective.

### 3.4. Natural Killer (NK)T Cells

After the presentation of lipid antigens, NKT cells immediately secret large amounts of cytokines including IFNγ and IL-4 to exhibit immunoprotective functions [51]. The fact that these cells are activated by lipid antigens makes it an attractive hypothesis to investigate their role in NAFLD, an impairment in fat homeostasis. Moreover, HSC have been confirmed to be professional APCs including the ability to present lipid antigens to NKT cells [42]. CD1d-restricted NKT cells are subdivided into two subsets namely Type I and II. During liver damage, Type I cells predominantly play pro-inflammatory roles while Type II cells suppress proinflammatory responses. Type I NKT cells are suggested to influence NAFLD in similar ways to CD8 T cells where they play detrimental roles in NASH progression. In mice fed a choline-deficient HFD, an increased infiltration of Type I NKT cells was observed in the liver. While there is an increased progression of NASH in the wild-type NAFLD model, Type I NKT cell-deficient (CD1d knockout) mice are protected from NASH progression [47]. Moreover, Type I NKT cell-deficient mice show reduced weight gain, lower plasma alanine transaminase (ALT) level, lower NAFLD activity score (NAS), reduced macrophages infiltration in the liver and downregulation of liver fibrosis markers including α-SMA, collagen type 1 alpha 1 (Col1a1), and collagen type 1 alpha 2 (Col1a2). On the other hand, Type II NKT cells are mainly involved in immunoregulatory functions due to their ability to suppress inflammation. However, it is mainly the Type I NKT cells that are rapidly activated and accumulated in the liver during liver injury and the effect of type II NKT cells is poorly understood [52]. Further studies are required to reveal their possible roles.

### 3.5. Mucosal-Associated Invariant T (MAIT) Cells

While the direct role of MAIT cells in NAFLD is still unknown, some studies have investigated changes in their phenotype and immunological function. However, these studies report ambivalent effects of MAIT cells in NAFLD. Li et al. detected an elevated frequency of MAIT cells in the liver during NAFLD with a positive correlation with NAFLD activity score and a protective effect was observed through the regulation of macrophage polarization [53]. Another study that detected a decreased MAIT cell frequency in blood and an increase in the liver of NAFLD patients demonstrated the opposite effects. MAIT cell-enriched mice (*V*α19 Tg) showed increased liver fibrosis and accumulation of hepatic fibrogenic cells, whereas MAIT cell-deficient mice (MR1 knockout) were relatively resistant [54]. Therefore, further studies are required to understand their roles in different progression stages and conditions of NAFLD.

## 4. T Cells and the Gut-Liver Axis

The liver is an organ with the nearest contact with the circulatory system that receives blood from the portal vein which highly exposes it to bacterial components and metabolites from the gut. Therefore, the hepatic immune system is substantially influenced by the gut-liver axis. The most studied mechanism connecting the hepatic immune response and the gut is the translocation of microbes and their components (Figure 2) known as microbe-associated molecular patterns (MAMPS) to the liver [55]. These are the sources for the antigens that can stimulate intrahepatic T cell subsets which substantially influences their functions in liver pathophysiology. The composition of gut microbiota is greatly altered during diet-induced NAFLD which makes it even a more important factor in hepatic T cell responses.

As components of the adaptive immune system, γδ T cells through their γδTCR can recognize bacterial lipid antigens presented by CD1d [56]. Considering their elevated infiltration of the liver in NAFLD conditions, their activation is a major factor for progression depending on the homeostasis of the gut microbiota. Similarly, the CD4 and CD8 cells which are predominantly conventional αβ T cells recognize microbial peptides presented by the MHC class I and II discussed above. Alteration of the gut microbiota determines the balance of translocation of the antigenic components thereby affecting the liver immune system. For example, dysbiosis caused by a hepatic fibrosis model in mice can reset the TCR immune repertoire (IR) and decrease their diversity that in turn reduces the number of different antigens they can recognize [57]. Additionally, the balance of endotoxins and exotoxins translocated to the liver can determine the immune response of the innate-like nature of some T cell subsets. The overall stimulation of the intrahepatic immune system is, therefore, directly influenced by the homeostasis of the gut microbiota. NKT cells and MAIT cells are the non-conventional T cells that can recognize bacterial lipid antigens presented by CD1d and metabolites produced during bacterial biosynthesis of vitamin B2 (Riboflavin) presented by MR1, respectively. The stimulation of liver tissue-resident MAIT cells is strictly correlated with the relative abundances of bacterial species capable of biosynthesizing the above metabolite ligands which are in turn dependent on the presence of functional *rib* genes in the species [58]. In murine models, the frequency, maturation and phenotype of NKT cells have also been confirmed to be greatly impacted by the gut microbiota [59]. In summary, the proliferation, recruitment, and phenotypes of all T cell subsets are directly connected to the condition of the gut-liver axis mainly through the translocation of microbial components.

## 5. NK Cells

NK cells are a major component of the innate immune cells and the first line of defense against pathogens affecting the host by direct cytotoxic mechanisms or modulation of other immune cells by secreting cytokines and chemokines [60]. NK cells in healthy adults develop exclusively in the bone marrow and include 5–20% of the total lymphocytes in the peripheral blood and approximately 30–55% of the total lymphocytes in the healthy liver [61,62,63]. Phenotypic characterization of NK cells shows CD56 expression and lack of CD3 expression. CD56 expression of NK cells is further divided by surface density: CD56^dim^ and CD56^bright^. However, the population ratio of CD56^dim^ to CD56^bright^ varies with the location, given the majority, almost 90% of CD56^dim^CD16^+^ account for circulating NK cells in peripheral blood while CD56^bright^CD16^−^ are primarily located in secondary lymphoid organs and tissues [64,65]. Based on their functionality, the CD56^dim^CD16^+^ phenotype mediates cytotoxicity, whereas CD56^bright^CD16^−^ has an immunomodulatory role by secretion of cytokines [66].

NK cells remain in a tolerogenic state under homeostasis with the assistance of inhibitory receptors (KIRs, NKG2) that recognize Human leukocyte antigens (HLA)-I on autologous cells. Viral infection or tumor infiltration fails to express or downregulate HLA-I, thus activating NK cells to perform cytotoxic functions mediated by activating receptors (NKp46, CD16, NKG2D) [67,68,69]. NK cells can be broadly characterized into two subsets: the regulatory CD56^brt^ (stage 4) that produces cytokines and chemokines upon stimulation and the cytotoxic CD56^dim^ (stage 5) which accounts for 90% of the total circulating NK cell population [70]. The cytotoxicity is mediated either by directed killing or through the release of perforin and granzyme B as in pre-formed granular proteins. These proteins can induce apoptosis in tumor or virus-infected cells [71]. CD56 subsets also differ by several other NK cell receptor expressions. CD16^+^ CD27^−^ expression is associated with cytotoxic subset CD56^dim^ which was upregulated at stage 5 during the maturity of NK cells [72,73].

### 5.1. Functional Competencies of NK Cells in NAFLD 

NK cells populations are relatively high in the liver depending upon the species. Humans and rats showed 30% to 50% while murine showed approximately 10% of total hepatic lymphocyte populations [74]. However, liver resident NK cells and conventional NK cells demonstrate substantial differences in progenitor origin, phenotype, gene expressions that require different transcription factors [75]. To differ the conventional NK cells from liver resident NK cells, unique subsets CD49a^+^CD49b^−^ were identified [75,76,77]. Compared with conventional NK cells, liver resident NK cells express a high level of TNF-related apoptosis-inducing ligands and granzymes or perforin and employ substantial cytotoxicity against activated hepatic stellate cells and tumors cells [78,79].

Till now, the function and regulation of NK cells in NAFLD is debated. As an innate immune organ, the liver is immune tolerant. However, in predisposing conditions, such as chronic infections or tumorigenesis, cross-talk between liver resident antigen-presenting cells and lymphocytes has been involved which plays a dual role of protective or pathogenic, depending upon the microenvironment [80]. Various studies have provided evidence that supports NK cell function in regulating liver fibrosis by killing directly activated hepatic stellate cells through the activating receptors NKG2D and NKp46 and the p38/PI3K/AKT pathway [81,82,83]. Other studies also suggest that during pathological conditions, NK cells accumulate in the liver and showed enhanced responsiveness against pathogens or tumors which can be beneficial in inhibiting disease progression [81,82]. This enhanced reactivity, in turn, can induce hepatocellular damage and possibly inhibit liver regeneration which may promote disease progression [84]. Therefore, regardless of the significant differences between liver-resident NK cells and conventional NK cells, it is still unknown whether these two subsets play a differential role in the pathogenesis of liver diseases.

In recent years, researchers have found that NK cells are increased in hepatic inflammation associated with NASH patients. In contrary to this, NK cells in NAFLD patients were reduced [85]. This decreased frequency was correlated with impaired functional activity leading to less granzyme/perforin production and interferon-γ, eventually reducing cytotoxicity and tumor-killing [11,86]. Interestingly some contradictory studies also indicate that NK cells potentially lose the cytotoxic activity during NASH which exacerbates in NASH patients to later stages of liver diseases [87,88]. While most of the studies are in support of NK cells’ recruitment to the liver during the NASH stage where they promote antifibrotic effect [11,74,81,82,83]. Despite such promising avenues in the field of NK cells aiming for NAFLD, care should be taken at the different stages of liver disease (Table 2).

### 5.2. NK Cells and Gut-Liver Axis

Apart from critically serving as the first line of defense against pathogens or abnormal cells by mediating cytotoxicity and recruiting other immune cells by secretion of cytokines, NK cells require priming for activation through various released cytokines in the gut [95]. Liver-resident NK cells also maintain tolerability and provide protection against viral infection or tumor and promote fibrosis regression by killing activated hepatic stellate cells [83,96]. Some of the previous studies stated that there was almost no difference in the cytotoxic functions of NK cells in germ-free and specific pathogen-free mice [97,98]. This observation was further confirmed by a study where Ganal et al. found out that commensal bacteria do not have any effect on the percentage or absolute NK cell numbers in the spleen and in accordance with this phenotypic change, splenic NK cells as well are not associated with commensal bacteria. Conversely, loss of NK cell activity in mice infected with cytomegalovirus and subsequent treatment with antibiotics and recolonization was reversed indicating that NK cell activity can be dynamically modulated by commensal bacteria [99]. Thus, providing insights into possible immunomodulatory NK cell responsiveness through the gut-liver axis.

Compelling evidence suggests that the composition of gut microbiota shapes the tolerability responsiveness of hepatic immune cells [100]. In light of this, studies are pointing to a role for the commensal microbiome in influencing the immune environment of liver disease and promotes different approaches, such as fecal microbial transplantation, probiotics, prebiotics, or bacterial consortium for compositional changes in microbiota, suggesting potential therapeutic benefits in modulating immune responses. Intriguingly, some studies showed that probiotics exert negative immunomodulation. In an in vitro study, Johansson et al., cocultured *Lactobacillus rhamnosus* GG (*L. rhamnosus* GG) and *Lactobacillus reuteri* DSM 17,938 with *Staphylococcus aureus* stimulated peripheral mononuclear cells. NK cells and T cells proliferated under the stimulus and produced IFN-γ, however, this lymphocyte activation was inhibited by the *L. rhamnosus* GG and *L. reuteri* suggesting the regression of microbial-induced immune activation. On the contrary, coculturing with *Lactobacillus plantarum* in IL-22 stimulated NK cells, exhibited high cytotoxic receptor expression. In addition to this, in another study feeding with *Bacillus subtilis* BS02 and BS04, enhanced NK cell cytotoxicity implying probiotics exhibit different functions under various factors and conditions in the diseased state.

Up to now, the importance of commensal bacteria in the regulation of hepatic NK cell functions remains largely unexplored. Host-microorganism interactions and their mechanism has been largely investigated by the use of modern analytical methods that have uncovered a critical role of gut microbiota in host health by generating metabolites that act both locally as well as systemically to influence human normal physiology and pathophysiology [101,102]. The most extensively studied metabolites are SCFAs, such as propionate, butyrate and acetate, primary and secondary bile acids and tryptophan-derived metabolites [103,104]. Additionally, with the advanced techniques, refining the biological functions pertaining to these metabolites is still ongoing. A valuable new insight relating to metabolic inter-communication between the host and gut microbiota, aryl hydrocarbon receptor (AHR) has caught researchers worldwide since many microbial metabolites can modulate AHR activity [105,106]. According to recent studies, tryptophan metabolites which are potent AHR ligands, have been found to activate AHR, subsequently reducing inflammation and increasing metabolic activity by downregulating de novo lipogenic genes thus ameliorating NAFLD [107,108]. Intriguingly, liver-specific knockout of AHR worsened the high-fat diet induced steatosis, and inflammatory responses in the liver simultaneously causing lipotoxicity to non-adipose tissues. The mechanism underlying bidirectional AHR-microbes communication is multi-faceted, involving intestinal barrier integrity and homeostasis [109,110], immune response modulations and tolerance [111], and carcinogenesis as well [112]. Because the gut and liver are so intertwined, given the anatomical location, AHR might be involved in the maturation and cytotoxic functions of NK cells in the liver. There is no direct evidence supporting the fact that tryptophan metabolites released from the gut could be parts of the key factors for NK cell maturation and cytotoxic activity. However, a study of innate lymphoid cells (ILCs) using AHR knockout mice suggested a distinct CD49a+TRAIL+CXCR6+DX5−NK1.1+ ILC/NK cell subtype residing in the liver which failed to perform its cytotoxic functions. From this study, it was anticipated that gut microbiota-derived AHR ligands might originally drive systematic immunity changes and subsequently affect liver resident ILC/NK cells [113].

## 6. Concluding Remarks and Future Perspectives

In this review, we discussed how different subsets of T cells and NK cells impact the hepatic immune system during the progression of NAFLD. It is now well established that hepatic immune response is directly related to the communication between the liver and gut during high-fat diet-induced liver impairment. However, the differential functional and phenotypical responses of T cells and NK cells depending on the type and progression stages of NAFLD are not well understood yet. Therefore, more information is required on the specific nature and signaling mechanisms of altered immune responses during liver diseases. A detailed understanding of the relationship between the dynamics of each immune cell type and particular ligands and metabolites from gut microbes, as well as the impaired liver microenvironment, should be established. This can be achieved through conducting genomic, transcriptomic and metabolomic exploration tools. This will provide further insights into NAFLD physiopathology and help identify therapeutic targets that can lead to designing translational immunotherapy.

## Figures and Tables

**Figure 1 ijms-22-12190-f001:**
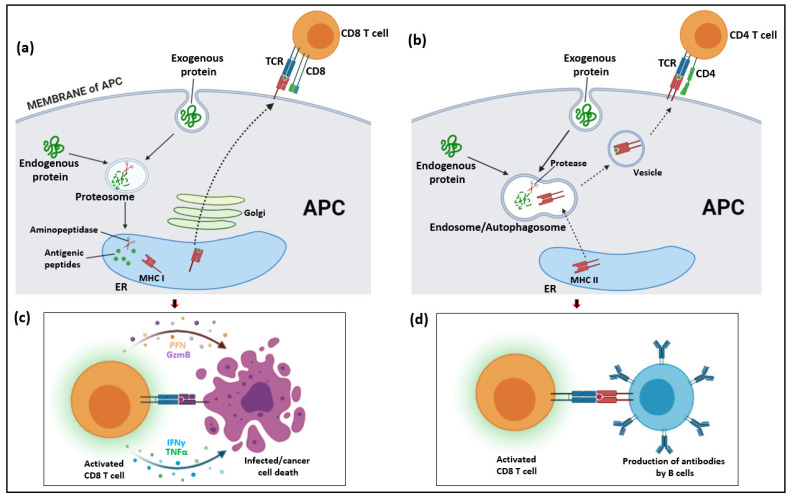
Mechanisms of antigen processing and T cell immune responses. (**a**) MHC class I antigen processing; (**b**) MHC class II antigen processing; (**c**) CD8 (Cytotoxic) T cell immune response; (**d**) CD4 (T helper) cell immune response.

**Figure 2 ijms-22-12190-f002:**
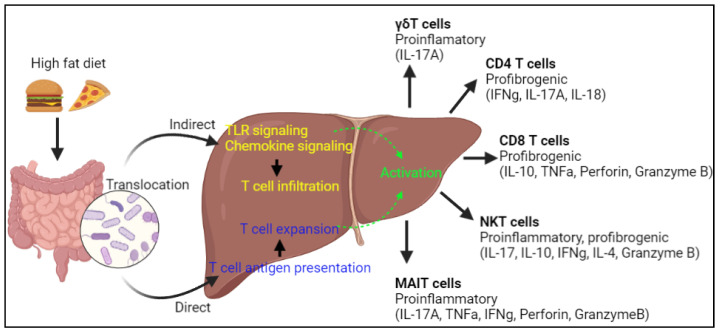
A simplified schematic representation of the gut-liver axis and T cell functions in fatty liver pathogenesis. IL, Interleukin; CD, Cluster of differentiation; TNF, Tumor necrosis factor; IFN, Interferon; NKT, Natural killer T.

**Table 1 ijms-22-12190-t001:** Summary of antigen processing in T cell subsets.

T cells	Subsets	TCR	Antigenpresentation	Main Antigen Molecules
Conventional	CD4 T cells	αβTCR	MHC-II	Peptides (Bacterial & host)
	CD8 T cells	αβTCR	MHC-I	Peptides (Bacterial & host)
Non-conventional	γδ T cells	γδTCR	MHC not necessary	(Bacterial & host)
	NKT cells	αβTCR	MHC-I like	Lipids & peptides (Bacterial & host)
	MAIT cells	αβTCR	MHC-I like	Vit-B2 metabolites (Bacterial)

CD: Cluster of differentiation, NK: Natural killer, TCR: T cell receptor, MHC: major histocompatibility complex.

**Table 2 ijms-22-12190-t002:** Summary of NK cell subsets in non-alcoholic fatty liver disease.

Studies	Major Findings	Activation/Cytotoxicity	References
Murine NASH	Increased CD49b+ NKp46+NK cells. They play a role in polarizing Mϕ toward M1-like phenotypes.	Immunoregulatory depends on IFN-γ, but not granzyme	[89]
Human NAFLD	Decreased frequency of CD56 (dim)NK cells and MAIT cells in PBMC	Less NKG2D	[90]
Human and murine NAFLD	Adipose tissue NK cells (or ILC1-like cells) contribute to insulin resistance in mice express CD49a. However, no link between the presence and levels of adipose tissue CD49a+ NK cells and the presence of insulin resistance was noted in the investigated patients.	Adipose CD49a+ mixed ILC1s expressed the most IFN-γ, the least granzyme B, and no TRAIL, unlike ILC1s in liver.	[91,92,93]
Human NAFLD	NAFLD with F3-F4 fibrosis scores exhibited elevated levels of circulating cytotoxic CD56(dim)CD16(+) cells	Inhibition of NK activity correlated with decreased expression of insulin receptors. mTOR/ERK inhibition correlates with decreased CD56 dim insulin receptor expression and NK impairment.	[94]
Human NAFLD	NK cells were shown to have an important role in regulating liver fibrosis by directly killing activated hepatic stellate cells via the receptors NKG2D, NKp30 and TRAIL the p38/PI3K/AKT pathway	NK cells are activated by both cytokines, such as IL-12 and IL-18, and innate immune stimuli, such as ligation of TLRs. The secretion of IL-18 depends upon activation of the inflammasome, whereas TLRs are stimulated by microbial products.	[83]

NAFLD, nonalcoholic fatty liver disease; NASH, nonalcoholic steatohepatitis; CD, co-receptor cluster of differentiation; NK, natural killer; ILC, innate lymphoid cells; MAIT, mucosal-associated invariant T; TLR, toll like receptor; IL, interleukin.

## Data Availability

Not applicable.

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
