# Peer review of "T Cell Subsets and Natural Killer Cells in the Pathogenesis of Nonalcoholic Fatty Liver Disease"

_ijms, 2021, doi:10.3390/ijms222212190_

Round 1
Reviewer 1 Report
This is an interesting review showing the roles of different subsets of T-cells and Natural killer cells in pathogenesis of nonalcoholic fatty liver disease. Although authors have tried to incorporate a lot of information in review, few things are lacking.
Comment 1: On Page 1, first few lines of introduction, authors have mentioned that NASH mainly related to high fat diet. It has been found that metabolic disorders including diabetes, thyroid, obstructive sleep apnea and polycystic ovary syndrome are also linked with pathogenesis of NASH. Authors should include these in Introduction.
Comment 2: On Page 5, in “CD8+ cytotoxic T cells” section, authors should include some more studies to show the role of CD8+ T-cells in NASH pathogenesis. One study showed that “CD8+ tissue-resident memory T cells promote liver fibrosis resolution by inducing apoptosis of hepatic stellate cells”. This indicates that CD8+ T-cells also have an important role on fibrosis resolution. One more study revealed that Perforin derived from CD8+ T-cells restricts hepatic inflammation in mice with NASH. These protective roles of CD8+ T-cells should be discussed in review.
Ref- 1. Koda Y, Teratani T, Chu PS. et al. CD8+ tissue-resident memory T cells promote liver fibrosis resolution by inducing apoptosis of hepatic stellate cells. Nat Commun 12, 4474 (2021). https://doi.org/10.1038/s41467-021-24734-0.
- Wang T, Sun G, Wang Y. et al. The immunoregulatory effects of CD8 T-cell-derived perforin on diet-induced nonalcoholic steatohepatitis. FASEB J. 2019 Jul;33(7):8490-8503. doi: 10.1096/fj.201802534RR.
Comment 3: Role of NKT cells has not been explained very well. Authors should add some literature to specify the role of NKT cells in pathogenesis of NASH. Additionally, now it is becoming clear that there are two different subsets of NKT cells—type I and type II and they have different modes of antigen recognition and have opposing roles in inflammatory liver diseases. Type I NKT cells predominantly have pro-inflammatory role in liver damage whereas type II NKT cells can suppress the pro-inflammatory response and protect against liver damage.
Also, authors mentioned that NKT cells generally recognize lipid antigens. It is well known that hepatic stellate cells are professional liver-resident antigen-presenting cells and present lipid antigens to NKT cells in CD1-restricted manner. This should be discussed in current review.
Ref- Winau F, Hegasy G, Weiskirchen R. et al. Ito cells are liver-resident antigen-presenting cells for activating T cell responses. Immunity. 2007 Jan;26(1):117-29. doi: 10.1016/j.immuni.2006.11.011.
Comment 4: In caption of Figure 1, authors have put MHC I antigen processing in both (a) and (b). Please correct it.
Comment 5: Authors have not discussed anything about regulatory T-cells (T-regs). Include one section stating the importance of T-regs in regulating the NASH pathogenesis.
Comment 6: There are minor spelling mistakes in manuscript such as on Page 6, line 227, though should be replaced by through. Please check for other spelling mistakes.
Author Response
ijms -1439934 \
“T Cell Subsets and Natural Killer Cells in the Pathogenesis of Nonalcoholic Fatty Liver Disease”
Point-to-point responses to comments by Reviewer 1
First, we would like to thank the reviewer for his/her comments, which helped us improve this manuscript. We then describe our point-to-point responses in the following table.
SN |
Reviewer’s Comments and suggestions |
Author’s responses |
-- |
This is an interesting review showing the roles of different subsets of T-cells and Natural killer cells in pathogenesis of nonalcoholic fatty liver disease. Although authors have tried to incorporate a lot of information in review, few things are lacking. |
ü We are grateful for your detailed review, positive comments and the critical issues you raised that helped us improve the manuscript. |
1 |
On Page 1, first few lines of introduction, authors have mentioned that NASH mainly related to high fat diet. It has been found that metabolic disorders including diabetes, thyroid, obstructive sleep apnea and polycystic ovary syndrome are also linked with pathogenesis of NASH. Authors should include these in Introduction. |
ü In agreement with the reviewer, the following content is added from line 32-36. ‘Moreover, existing data support that type 2 diabetes (T2D), hyperthyroidism and obstruc-tive sleep apnea are important contributing factors for the progression of NAFLD [1-3]. A recent retrospective cohort study of women diagnosed with polycystic ovarian syndrome (PCOS) also reported that PCOS is a risk factor associated with subsequent NAFLD occur-rence [4].
|
2 |
On Page 5, in “CD8+ cytotoxic T cells” section, authors should include some more studies to show the role of CD8+ T-cells in NASH pathogenesis. One study showed that “CD8+ tissue-resident memory T cells promote liver fibrosis resolution by inducing apoptosis of hepatic stellate cells”. This indicates that CD8+ T-cells also have an important role on fibrosis resolution. One more study revealed that Perforin derived from CD8+ T-cells restricts hepatic inflammation in mice with NASH. These protective roles of CD8+ T-cells should be discussed in review. Ref- 1. Koda Y, Teratani T, Chu PS. et al. CD8+ tissue-resident memory T cells promote liver fibrosis resolution by inducing apoptosis of hepatic stellate cells. Nat Commun 12, 4474 (2021). https://doi.org/10.1038/s41467-021-24734-0. 2. Wang T, Sun G, Wang Y. et al. The immunoregulatory effects of CD8 T-cell-derived perforin on diet-induced nonalcoholic steatohepatitis. FASEB J. 2019 Jul;33(7):8490-8503. doi: 10.1096/fj.201802534RR. |
ü We thank the reviewer for pointing this out and recommending contents and references. Accordingly, the following content has been added from line 214-220. ‘Moreover, CD8+ tissue-resident memory T cells have been reported to promote liver fibro-sis resolution by inducing apoptosis of HSC indicating their important roles in fibrosis resolution [49]. On the contrary, cytotoxic T cells have been implicated for exerting protective effects where they were able to restrict hepatic inflammation in mice with NASH ac-companied by decreased plasma levels of liver enzymes. Therefore, further exploration is required to reveal the differential roles of CD8+ cytotoxic T cells in different conditions of NAFLD [50]. Additionally.’ |
3 |
Role of NKT cells has not been explained very well. Authors should add some literature to specify the role of NKT cells in pathogenesis of NASH. Additionally, now it is becoming clear that there are two different subsets of NKT cells—type I and type II and they have different modes of antigen recognition and have opposing roles in inflammatory liver diseases. Type I NKT cells predominantly have pro-inflammatory role in liver damage whereas type II NKT cells can suppress the pro-inflammatory response and protect against liver damage. Also, authors mentioned that NKT cells generally recognize lipid antigens. It is well known that hepatic stellate cells are professional liver-resident antigen-presenting cells and present lipid antigens to NKT cells in CD1-restricted manner. This should be discussed in current review. Ref- Winau F, Hegasy G, Weiskirchen R. et al. Ito cells are liver-resident antigen-presenting cells for activating T cell responses. Immunity. 2007 Jan;26(1):117-29. doi: 10.1016/j.immuni.2006.11.011. |
ü In agreement with the reviewer, we have elaborated the discussion about NKT cells and revised the section as follows from line 225-245. ‘After the presentation of lipid antigens, NKT cells immediately secret large amounts of cytokines including IFNγ and IL-4 to exhibit immunoprotective functions [51]. The fact that these cells are activated by lipid antigens makes it an attractive hypothesis to investigate their role in NAFLD, an impairment in fat homeostasis. Moreover, HSC have been confirmed to be professional APCs including the ability to present lipid antigens to NKT cells [42]. The CD1d-restricted NKT cells are subdivided into two subsets namely Type I and II. During liver damage, Type I cells predominantly play pro-inflammatory roles while Type II cells suppress proinflammatory responses. Type I NKT cells are suggested to influence NAFLD in similar ways as CD8 T cells where they play detrimental roles in NASH progression. In mice fed choline deficient HFD, an increased infiltration of Type I NKT cells is observed in the liver. While there is an increased progression of NASH in wild type NAFLD model, Type I NKT cell deficient (CD1d knockout) mice are protected from NASH progression [47]. Moreover, Type I NKT cell deficient mice show reduced weight gain, lower plasma alanine transaminase (ALT) level, lower NAFLD activity score (NAS), reduced macrophages infiltration in the liver and downregulation of liver fibrosis markers including α‐SMA, collagen type 1 alpha 1 (Col1a1), and collagen type 1 alpha 2 (Col1a2). On the other hand, Type II NKT cells are mainly involved in immunoregulatory functions due to their ability to suppress inflammation. However, it is mainly the Type I NKT cells that are rapidly activated and accumulated in the liver during liver injury and the effect of type II NKT cells is poorly understood [52]. Further studies are required to re-veal their possible roles. ‘ |
4 |
In caption of Figure 1, authors have put MHC I antigen processing in both (a) and (b). Please correct it. |
ü Error in the caption has been corrected in line 104. |
5 |
Authors have not discussed anything about regulatory T-cells (T-regs). Include one section stating the importance of T-regs in regulating the NASH pathogenesis. |
ü We are grateful for the reviewer for the suggestion. In agreement with the suggestion, we have divided the CD4+ T cells into two sections and included the following content as an additional subsection under the CD4 T cells from line 183-198. ‘Another subset of T cells which account for 5-10% of the total number of CD4+ T cells in healthy humans are regulatory T cells (Tregs). As a T cell subpopulation, they are produced in the thymus and express the transcription factor forkhead box P3 (Foxp3) of the forkhead family [39]. Their importance in the development and pathogenesis of NAFLD is poorly recognized. While little is known about the significance of intrahepatic Tregs in NASH, it is well established that they support the homeostasis of immune response by inhibiting proliferation and activation of cytotoxic T cells [40, 41]. Tregs have been con-firmed to be involved in prevention of autoreactive cells proliferation in autoimmune hep-atitis, negative control of immune response in NASH and organ tolerance during transplanatation [42]. Induction of Tregs has been demonstrated to alleviate insulin resistance thereby contributing to amelioration of liver damage. A study that investigated the effect of inducing Tregs using the immunosuppressant OKT3 in NASH patients found that im-mune parameters and liver function was improved [43]. Another recent study that found an increased intrahepatic Tregs in mice NASH models also reported their detrimental roles [44]. However, further studies are required to elucidate the detailed mechanisms for their roles which can help their applications in immunotherapy of NASH.’ |
6 |
There are minor spelling mistakes in manuscript such as on Page 6, line 227, though should be replaced by through. Please check for other spelling mistakes. |
ü Several spelling errors have been corrected at several parts of the manuscript. |

Reviewer 2 Report
The authors explore the relevance of different T cells subsets and NK cells in NAFLD as critical immune players in and during NAFLD progression. The review is comprehensive, well structured, and the schematic figures support the text.
I have the following comments for the authors to address
- The authors need to include a sentence for NAFLD in the abstract.
- The authors need to acknowledge the gap in the knowledge in the Introduction section.
- Introduction/line46-47/T cells and NK cells.../ The authors need to reference relevant literature
- Introduction/line 54/correct "induc4es" to "induces"
- Figure 1A: Indicate in labeling "Endogenous protein"
- CD4+ T helper cells/ Lines 166-167: The authors should include relevant literature.
- NK cells and gut-liver axis/line 366/change "metabolites releasing" to "metabolite released."
Author Response
ijms -1439934 \
“T Cell Subsets and Natural Killer Cells in the Pathogenesis of Nonalcoholic Fatty Liver Disease”
Point-to-point responses to comments by Reviewer 2
First, we would like to thank the reviewer for his/her comments, which helped us improve this manuscript. We then describe our point-to-point responses in the following table.
SN |
Reviewer’s Comments and suggestions |
Author’s responses |
-- |
The authors explore the relevance of different T cells subsets and NK cells in NAFLD as critical immune players in and during NAFLD progression. The review is comprehensive, well structured, and the schematic figures support the text. I have the following comments for the authors to address |
ü We thank the reviewer for the valuable comments and recommendations that helped us improve the manuscript. |
1 |
The authors need to include a sentence for NAFLD in the abstract. |
ü In agreement with the reviewer, we have added a sentence that mentions NAFLD as follows in line 13-14. ‘Nonalcoholic fatty liver disease (NAFLD) is a condition characterized by hepatic accumulation of excess lipids.’ |
2 |
The authors need to acknowledge the gap in the knowledge in the Introduction section. |
ü According to the reviewer’s suggestion, the following content is added in lines 69-74. ‘While association of the most common classes of immune cells with several cardiovascular and autoimmune diseases is relatively well studied, the ambivalent roles of different subsets of intrahepatic T cells and NK cells remain unexplained.’ |
3 |
Introduction/line46-47/T cells and NK cells.../ The authors need to reference relevant literature |
ü The following references are cited in line 52-53. References [11, 12] |
4 |
Introduction/line 54/correct "induc4es" to "induces" |
ü Spelling error is corrected in line 59. |
5 |
Figure 1A: Indicate in labeling "Endogenous protein" |
ü We thank the reviewer for noticing. The label has been added in figure 1A. |
6 |
CD4+ T helper cells/ Lines 166-167: The authors should include relevant literature. |
ü This content was included based on a data from the following article and the following citation is inserted in line 174. Reference [33] |
7 |
NK cells and gut-liver axis/line 366/change "metabolites releasing" to "metabolite released." |
ü Grammar error has been corrected in line 411-412. |
